# Production of Itraconazole Nanocrystal-Based Polymeric Film Formulations for Immediate Drug Release

**DOI:** 10.3390/pharmaceutics12100960

**Published:** 2020-10-13

**Authors:** Anna Karagianni, Leena Peltonen

**Affiliations:** 1Division of Pharmaceutical Technology, Aristotle University of Thessaloniki, 54124 Thessaloniki, Greece; karagiak@pharm.auth.gr; 2Drug Research Program, Division of Pharmaceutical Chemistry and Technology, Faculty of Pharmacy, University of Helsinki, P.O. Box 56, Viikinkaari 5 E, 00014 Helsinki, Finland

**Keywords:** film casting, improved solubility, media milling, nanocrystals, polymeric film formulation

## Abstract

In order to improve the solubility properties of BCS class II drug itraconazole, fast dissolving oral polymeric film formulations based on itraconazole nanocrystals were produced. Drug nanocrystals were manufactured by the wet pearl milling technique. In polymeric film formulations, hydroxypropyl methyl cellulose (HPMC) was used as a film forming polymer, and glycerin was used as a plasticizer. For nanocrystal suspensions and film formulations, thorough physicochemical characterization was performed, including particle sizing and size deviation, film appearance, weight variation, thickness, folding endurance, drug content uniformity, disintegration time, and dissolution profile. After milling, the nanoparticles were 369 nm in size with a PI value of 0.20. Nanoparticles were stable and after redispersion from film formulations, the particle size remained almost the same (330 nm and PI 0.16). The produced films were flexible, homogeneous, fast disintegrating, and drug release rate from both the nanosuspension and film formulations showed immediate release behavior. Based on the study, the film casting method for production of itraconazole nanocrystal based immediate release formulations is a good option for improved solubility.

## 1. Introduction

Decreasing the particle size is a very efficient way to improve the solubility of poorly soluble drug materials [1,2]. However, formation of nanocrystals is just one way to manipulate the raw material, and after nanonization, the final formulation still needs to be made. Nanosuspensions can be used, for ocular drug delivery [3], for example, but, for solid dosage forms, the formulation process is typically a multistage process. Nanosuspensions are formulated to solid dosage forms, like tablets, capsules, granules [4], or more recently, oral polymeric films [5]. The benefits of oral polymeric films (also called oral thin films or oral strips) are following—they can be administered without water, they are patient-friendly, and they can be utilized for personalized medicine purposes [6,7]. For nanocrystal-based drug delivery systems, polymeric films are especially beneficial, while no extra drying step is required—nanosuspensions can be added as such to the polymeric solution for the film casting or 3D printing.

Oral polymeric films can be produced, for example, by solvent casting [5], hot melt extrusion [8], 3D printing [9], or electrospinning technologies [10]. Typically, oral polymeric films contain at least one film-forming polymer (like starch, PVP, HPMC) and a plasticizer (like glycerol, propylene glycol). The presence of plasticizer decreases the brittleness of the film and makes it more durable. Film formulations might also contain sweeteners and flavors. The active pharmaceutical ingredients (APIs) delivered via oral polymeric films are mostly swallowed with saliva to the GI tract, and the absorption takes place in the GI tract. Part of the APIs can also be absorbed through oral mucosa.

Oral polymeric film formulations are a good option for water soluble active pharmaceutical ingredients (APIs), but recently they were also studied for poorly soluble drug applications. For example, in a study by Shen et al. [11], amorphous antiviral herpetrione nanoparticles were loaded into an oromucosal film formulations. Mixture of hydroxypropyl methyl cellulose (HPMC), hydroxypropyl cellulose (HPC), microcrystalline cellulose (MCC), and polyethylene glycol (PEG) was used as film-forming polymers, and nanoparticle-loaded films were produced via the film casting technique. The formulated films disintegrated within 30 s and the dissolution was fast [11]. However, the production of film formulations with (nano)particle dispersions is challenging, due to the difficulty in reaching high homogeneity and because of the mechanical properties of the films [12].

Itraconazole is a poorly soluble Biopharmaceutics Classification System (BCS) class II drug. Optimum dissolution area for itraconazole is in the stomach, while in the intestine, its solubility is approximately 250 times less, as compared to the low pH value in stomach [13]. With fast dissolving systems, this might raise problems due to the uncontrolled precipitation/crystallization in vivo, as was shown in our earlier studies with itraconazole nanocrystal formulations [4,14]. The bioavailability of itraconazole was increased when the nanocrystals were impeded into a nanocellulose matrix structure [14]. However, if the nanocrystals were freeze-dried and packed into capsules, the bioavailability was lower, due to the fast transition of the dissolved drug into the intestine, where the already dissolved itraconazole was prone to precipitate, due to the lower solubility [4]. Certain well-known stabilizing polymers (for example, HPMC) was shown to be beneficial for maintaining a supersaturated state and for avoiding precipitation [15].

In this study, we aimed for production of immediate release formulation, based on drug nanocrystals. Polymeric films were selected as the formulation due to their simple structure, good performance, and fast production. Further, functionality was built into the formulation, by using HPMC as a film former—HPMC stabilized the supersaturated state, which was reached after the dissolution of drug nanocrystals. Presence of HPMC hindered uncontrolled precipitation, which is often induced by supersaturation. Accordingly, in this study, itraconazole nanocrystal-based oral polymeric film formulations for immediate release purposes were produced. The film formulations were made by the film casting method. HPMC was used as a film-forming polymer. For film casting, the drug was nanomilled to nanocrystals. After the production of nanocrystal-based polymeric film formulations, a thorough physicochemical and mechanical analysis (particle size and size deviation, film appearance, weight variation, thickness, folding endurance, drug content uniformity, disintegration time, and dissolution profile) were performed to show their suitability for immediate drug release purposes.

## 2. Materials and Methods

### 2.1. Materials

Itraconazole (Orion Pharma, Espoo, Finland) was used as a model drug. Poloxamer 407 (Lutrol F127, BASF Co., Ludwigshafen, Germany) was used as a stabilizer for drug nanocrystals. Hydroxypropyl methyl cellulose (HPMC E5 Premium LV, Dow Chemical Company, Midland, MI, USA) was used as the film former in the film casting, and glycerol 85% (Oriola, Espoo, Finland) was used as a plasticizer in the film forming. Water used throughout the study was ultrapurified Milli-Q-water (Millipore SAS, Molsheim, France).

### 2.2. Nanocrystallization by Wet Milling

Itraconazole nanosuspensions were produced by the wet milling method in a planetary ball mill (Pulverisette 7 Premium, Fritsch Co., Idar-Iberstein, Germany), using Poloxamer F127 as a stabilizer. Method optimization was based on our earlier studies [16]. The milling vessel was filled with milling pearls, drug, and aqueous stabilizer solution. The parameters for the milling were following—40 mL milling vessel (zirconium oxide) including 70 g of milling pearls (Ø1 mm, zirconium oxide) and drug/stabilizer suspension (itraconazole 2 g; poloxamer F127 0.80 g; water 10 mL). The suspension was milled with a rotational speed of 1100 rpm for 3 min, after which the vessel was cooled down for 10 min time in an ice bath. Three minute milling and 10 min cooling was repeated 4 times, the total milling time being 12 min. For film casting, the nanosuspension was used as such.

### 2.3. Particle Size and Polydispersity Index

Particle size and polydispersity index (PI) analysis for the itraconazole nanosuspensions were performed by photon correlation spectroscopy (PCS) (Malvern Zetasizer 3000HS, Malvern Instruments, Malvern, UK). PI indicates the width of the particle size distribution; the lower the PI, the more monodisperse and uniform in size are the particles. If the PI value is above 0.7, the system is considered to be polydisperse. Measurements were performed both from the freshly prepared nanocrystal suspensions and after the redispersion of the nanocrystals from the polymer films. For the redispersion, a small part of a film was added to 3 mL of water and sonicated for 3 min. All measurements were performed three times.

### 2.4. Film Casting

For film casting, the film-forming polymer was wetted with the plasticizer (glycerol), after which the water was added. The amount of HPMC varied from 2.5 g to 5.3 g (in 100 mL total volume). The amount of glycerol in all studies was 25% (*w/w*), as compared to the amount of HPMC (the exact compositions of the films are presented in Section 3.1 and Section 3.2). In order to get a homogeneous viscous solution, the polymer was grinded properly and put under a magnetic stirring (IKA, Staufen, Germany) for 4 h. After the stirring, the solution was left to rest, in order to remove the air bubbles. For film casting, the drug nanosuspension was diluted with water and mixed with the HPMC solution, and sonicated for 6 min, in order to reach a homogenous system. The solution was kept overnight in a fridge, in order to remove the air bubbles. For film casting, the solution (9 mL) was casted to a petridish, after which it was dried for 14 h in an oven at 41 ± 1 °C, for solvent evaporation and dry film formation. For the blank reference polymer films, water was used instead of the nanoparticle suspension.

The polymeric film formulations were characterized in terms of their appearance, weight variation, thickness, folding endurance, drug content uniformity, disintegration time, and dissolution profile.

### 2.5. Appearance of the Films

The appearance of the films was checked visually. Acceptable films should have a smooth, soft, and flexible appearance, without folding and bubbles. The blank films should be transparent with high clarity.

### 2.6. Folding Endurance

The folding endurance is an indicator for the film brittleness and it is expressed as a numerical value describing the number of folds required for the film to break (or to develop visible split). Folding was repeated for the same specified film area as long as the film cracks were detected. The test was run 5 times.

### 2.7. Weight Variation

For weight variation analysis, films were cut into small pieces (0.9 × 0.9 cm^2^, including 1.2 mg of drug). Three individual pieces of films were randomly selected and weighted and the average weight was calculated.

### 2.8. Film Thickness

For film thickness determinations, the thickness was measured by a digital micrometer (Sony Magnescale Inc., DZ521, Mie, Japan) from 5 different positions of each film (both central and edge positions).

### 2.9. Content Uniformity

For content uniformity test, the film was cut down to small pieces in the same way as for weight variation analysis (0.9 × 0.9 cm^2^, including 1.2 mg of drug). Each piece was placed in a volumetric flask with 50 mL of methanol, in order to dissolve the drug, and the system was stirred with a magnetic stirrer overnight, in order to achieve complete dissolution of all drug content. The concentration was measured by UV–Vis spectrophotometer (UV-1600PC, VWR Int., Leuven, Belgium) at 258 nm wavelength. Measurements were performed five times.

### 2.10. Disintegration

Disintegration test was performed through two different methods.

First, one 0.9 × 0.9 cm^2^ sized piece of film was put in a petridish containing 20 mL of water, and it was stirred every 10 s. The time required for the film to disintegrate was inspected visually. For each formulation, the test was performed for 6 times.

In the second test, the disintegration times were determined in a traditional basket-rack apparatus (Ph. Eur., Sotax AG DT3, Aesch, Switzerland). One piece of the film (0.9 × 0.9 cm^2^) was put in each tube of the basket (no discs were used). The basket was placed in a 1 L beaker filled with distilled water at 37 ± 2 °C. The time required for the film to break and dissolve was counted as the average of 6 samples. Disintegration time was defined to be the time where no clear film residues were seen.

### 2.11. Drug Release

Drug release studies (bulk drug, drug nanosuspensions, and nanocrystal-loaded films) were performed using the official paddle apparatus (Ph. Eur., Erweka DT-D6, Erweka, Heusentamm, Germany). In all tests, the drug amount in the test vessel was 1.2 mg. Dissolution tests were run for 1 h. As a dissolution medium, 1 L of 0.1 M hydrochloric acid was used, and the temperature during the test was kept at 37 ± 0.5 °C. Rotational speed was 100 rpm. At preselected time intervals (30 s, 1, 1.5, 2, 3, 5, 15, 30, and 60 min) 3 mL samples were taken, and they were replaced with the same volume of the fresh medium. The concentrations were determined by the UV–Vis spectrophotometer at 258 nm wavelength (filtered with 0.45-μm PVDF filter, before analysis). For dissolution analysis, six parallel samples were analyzed.

## 3. Results

### 3.1. Blank Polymer Films

For polymer films, HPMC was selected as a film former polymer, while it had good film-forming ability. It also had the ability to maintain and stabilize the supersaturated state of the dissolved drug [15]. Glycerol was used as a plasticizer and humectant in the composition. The studied concentration levels for HPMC and glycerol for the screening studies were selected, based on a literature survey [17,18,19,20]. The selected compositions for the study are presented in Table 1.

After production of the blank films, their appearance and mechanical properties were analyzed (Table 2).

All films had a high uniformity in their structure. The higher polymer concentrations resulted in increased film thicknesses. The average thickness value for the Batch 3 films (the 2nd highest amount of polymer) was 0.124 mm, whereas for the Batch 1 films (lowest polymer concentration) it was 0.087 mm. Increased polymer concentration produced films with higher roughness and lower flexibility.

From the four studied compositions, Batch 1 had the best film-forming properties, and it was selected for the drug loading studies.

### 3.2. Drug Loaded Films

For the drug loading tests, the HPMC concentration in the casting solution was selected to be 2.5% (*w/w*). For film casting, HPMC-glycerol solution (6 mL) was mixed with nanosuspension (3 mL), keeping the polymer solution—nanosuspension ratio 2:1.

First, 0.5 g of the original itraconazole nanosuspension was directly mixed with the polymer solution at the 1:2 ratio, but the obtained films were opaque and very brittle. Accordingly, two different approaches were used for optimization—(1) nanosuspension dilution in water, and (2) higher concentration of HPMC solution (Table 3). This meant that besides the optimum HPMC concentration found from the blank film tests (2.5% HPMC), as a reference sample, also one higher HPMC concentration (5.3% HPMC) was tested.

In accordance with the blank films, the films with lower polymer concentration were thinner and the drug was dispersed to the polymer matrices in a better way. Additionally, the lower HPMC concentration films were visually more homogenous and had a better appearance, while a higher polymer concentration made the films stiffer and less flexible. A high polymer concentration also complicated the mixing of the polymer solution with nanosuspension, which also lowered the homogeneity of the final film formulation.

### 3.3. Physcochemical and Mechanical Characterization of the Drug Loaded Films

Thorough physicochemical characterization was done for the best drug loaded film composition, Batch A (Table 4).

Itraconazole nanosuspension was effectively produced with poloxamer F127, through the wet milling technique, based on our earlier optimized system [16]. The particle size after the milling was 368.3 nm (PI value 0.20), while the corresponding particle size values for the itraconazole nanoparticles after redispersion of the films were 329.6 nm (PI 0.16).

In the morphological study, the films showed good appearance, they were smooth, non-sticky and flexible, and no folding or gaps/drug free areas were detectable. The folding endurance was quite high, indicating a very flexible and durable film.

Film thicknesses were approximately 0.1 mm, with low variation. Nanosuspensions were evenly distributed to the polymer matrix inside the film, and the content uniformity was in good level. In the pharmacopoeia disintegration test, the average disintegration times were 1.51 min. In petridish disintegration test, the average disintegration times were 2.53 min.

### 3.4. Drug Release Studies

In the drug release testing, film loaded nanoparticle formulations were compared to freshly milled drug nanosuspensions and bulk itraconazole. In all dissolution tests, the sample size corresponded to 1.2 mg of pure itraconazole.

As expected, bulk itraconazole had a very low drug release rate (Figure 1). Even after a 1 h dissolution time, less than 15% of the drug was dissolved.

On the other hand, both nanoformulations (nanosuspension, and film formulation) showed almost immediate drug release, starting from the first minutes of the testing. More than 50% of the nanosized itraconazole was dissolved after the first 2 min time, from both nanoformulations.

## 4. Discussion

In order to formulate fast disintegrating itraconazole nanocrystal-based films, first, blank polymer films were studied. HPMC is well-known for its good film forming ability, and in an earlier study, drug release from HPMC films were found to be fast, due to low viscosity [21]. Especially, the HPMC E5 LV polymer was shown to possess preferable film forming properties [22] and to have a beneficial impact on nanoparticle redispersibility [20]. Accordingly, HPMC E5 LV was selected as a film forming polymer for this study. HPMC E5 also had an optimal Tg value for oral film formulations, and HPMC is known to act as a stabilizer for the supersaturated state, which can be reached after nanocrystal dissolution [15].

Films containing high concentrations of HPMC do not have either good appearance or uniformity characteristics, and a longer time is needed to prepare a transparent, uniformly dissolved bubble-free polymer solution [23]. Additionally, high polymer concentration decreases the drug release rate [24]. Glycerol was selected to be used as a plasticizer and humectant in the composition, and based on the earlier studies, its amount was selected to be 25% (*w/w*), as compared to the amount of HPMC.

All blank films had a high structural uniformity. Polymer concentration was directly related to film thickness—high polymer concentration resulted thicker films. However, high polymer concentrations also produced rougher and less flexible films.

For the selection of the nanosuspension concentration in the film casting mixture, it was noticed that higher drug loadings generally required longer dissolution times for total drug release and, if the amount of the drug particles in the film was above 30% (*w/w*), the film tended to be brittle [5,12,25]. Poloxamers’ impact on the HPMC film structure was assumed to be a result of HPMC–poloxamer interaction, which results in a higher resistance in the gel, retards the erosion, and improves the film-forming ability [26].

Similar to blank films, the drug-loaded films with lower polymer concentration were thinner and the drug was well dispersed to the polymer matrix. Additionally, films with a lower polymer concentration were visually more homogenous and they had a better appearance. A higher polymer amount led to stiffer and less flexible films. The same kind of behavior was also found in an earlier study, where the film thickness was further correlated with the dissolution rate [27].

Itraconazole nanocrystal structures were not destroyed during the film casting or drying process and they maintained their original nanoparticle properties. The nanoparticles were fully recovered upon redispersion in water from the films, indicating that the film casting process was successful, and did not cause irreversible aggregation. The polydispersity indices, both immediately after the milling and after the redispersion of the film, were very low (below 0.2), indicating good stability and monodispersed particle sizes.

The dried films had a good appearance—they were smooth, non-sticky, and flexible, and no folding, gaps, or drug-free areas were detected. In film casting, the drying process is very gentle, and no aggregation or wrinkling of the surfaces were noticed, which are typical for more harsh drying conditions [28,29]. The high folding endurance, indicated good flexibility and durable film structures.

Film thicknesses were around 0.1 mm, with low variation. Nanoparticles were evenly distributed to the polymer matrix inside the film, and the content uniformity was at a good level—the determined drug amount in the film pieces was 1.25 ± 0.05 mg, while the theoretically calculated value was 1.2 mg.

Disintegration was tested with two different methods—Pharmacopoeia (Ph. Eur), and shaking on petridish. In both tests, films were fast to disintegrate—the disintegration times were below 3 min. In the pharmacopoeia method, disintegration times were 1.5 min. In this method, the films might be attached to the walls of the vessel, which might disturb the determinations. Additionally, in this method, small film pieces might be floating, or visible detection might be difficult due to the tube walls. For this reason, a simple disintegration test on a petridish was also performed, where the visual inspection of the disintegration was more reliable for thin-film systems. In this test, the average disintegration times were 2.5 min. The mixing in this method was not as efficient as in the pharmacopoeia method, which explained the longer (1 min) disintegration times.

The drug dissolution from the films was almost as fast as from the nanosuspensions, indicating a good performance of the film formulations. The fast and complete drug release from itraconazole nanosuspensions was mainly due to the increased surface-to-volume ratio. Fast drug release of itraconazole from the films was attributed to the high surface area of drug nanocrystals available for wetting, after fast disintegration of the water-soluble HPMC films. HPMC had shown in earlier studies to be able to maintain the supersaturated state longer [15]. However, in this study the dissolution tests were performed under sink conditions, which meant that at the end point after the dissolution, no supersaturated state was reached.

In an earlier study, it was shown that when the films were thicker the dissolution rate had slowed down [27]. As already shown in the redispersion tests, after the fast disintegration of the films, the nanoparticles were dispersed to the dissolution media with the original particle size, and enabled the immediate drug release—over 65% of the drug was released in 5 min. In this study, the drug release and film disintegration was fast, as compared to an earlier tadalafil nanocrystal-based oral film formulation [5]. In the study, tadalafil nanocrystals were considerably larger in size (from 588 nm to 945 nm) with higher polydispersity values. Tadalafil formulations disintegrated slower (4.04 min to 14.57 min), which also partly, besides the larger particle size, affected the slower dissolution.

Based on the redispersibility studies, itraconazole nanocrystals were not irreversibly aggregated in the film formulations. For nanoparticle formulations, it is common that aggregation takes place, which slows down the dissolution and leads to a lower bioavailability. This was shown with the nanoparticle-based formulations, where the particle aggregation hindered the possible dissolution enhancement, due to the nanostructures [28,30]. The aggregation tendency depended on the surface properties of the particles. The dissolution profile of the loaded films with the nanocrystals was related to the redispersibility properties of the nanoparticles.

As a conclusion, in this study, fast dissolving films of itraconazole nanoparticles were easily produced and in a satisfactory, cost-effective way. The film formulations showed good appearance, favorable redispersibility of the drug nanocrystals from the films, high homogeneity of films, adequate physical characteristics like smooth surface structure with good flexibility, favorable disintegration time, and immediate drug release profiles. Fast dissolving films made from biocompatible HPMC and loaded with poorly soluble drug nanocrystals, are an advantageous drug delivery form for patient-compliant drug therapy.

## Figures and Tables

**Figure 1 pharmaceutics-12-00960-f001:**
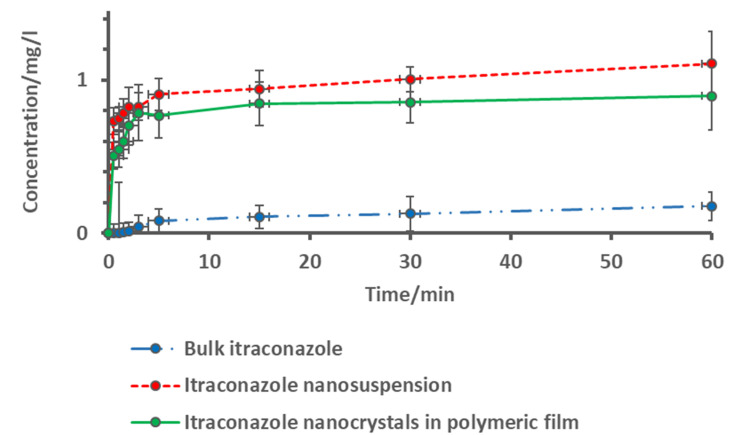
Drug release studies from bulk itraconazole, itraconazole nanosuspension, and itraconazole nanocrystal-based film formulation.

**Table 1 pharmaceutics-12-00960-t001:** Compositions for the studied blank (drug-free) films.

Batch	HPMC (g)	Glycerol (g)	Water (g)
1	2.5	1.0	96.5
2	2.8	1.1	96.1
3	3.0	1.2	95.8
4	3.1	1.3	95.6

**Table 2 pharmaceutics-12-00960-t002:** Film forming ability, transparency, and flexibility of the studied blank films.

Batch	Final HPMC Concentration in Casting Solution (% *w*/*w*)	Film Forming Ability	Appearance	Flexibility
1	2.5	Very good	Transparent	Flexible
2	2.8	Good	Transparent	Flexible
3	3.0	Average	Transparent	Semiflexible
4	3.1	Average	Transparent	Semiflexible

**Table 3 pharmaceutics-12-00960-t003:** Compositions of the drug loaded films.

Material	Batch A	Batch B	Batch C	Batch D
HPMC (g)	2.5	2.5	5.3	5.3
Glycerol (g)	1.0	1.0	2.13	2.13
Nanosuspension (mL)	2.76	5.7	2.76	5.7
Water ad (mL)	100	100	100	100

**Table 4 pharmaceutics-12-00960-t004:** Mechanical and physicochemical parameters for the drug-loaded films.

Test	Result
Folding endurance	52 ± 3
Weight variation (mg)	10.96 ± 0.36
Thickness (mm)	0.101 ± 0.004
Drug content uniformity (mg)	1.25 ± 0.05
Disintegration time (Pharmacopoeia test) (min)	1.51 ± 0.03
Disintegration time (Test on Petri dish) (min)	2.53 ± 0.23

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
