# Peer review of "Production of Itraconazole Nanocrystal-Based Polymeric Film Formulations for Immediate Drug Release"

_pharmaceutics, 2020, doi:10.3390/pharmaceutics12100960_

Round 1
Reviewer 1 Report
This is well written paper describing development of oral films for improving dissolution rate and bioavailability of poorly soluble drug itraconazole. These are suggestions for authors to improve the quality of the paper.
- It will be useful for reader to see photos and/or polarizing micrographs of prepared films (if films are sufficiently transparent) instead of only description of film appearance. Microscopy should proved absence of macroscopic aggregates of itraconazole crystals.
- Did authors used disks in pharmacopoeial disintegration apparatus? It is reasonable to expect that without disks films will floated out of the cylinders during test, particularly if the vessel is filled with medium, as described in this study.
- For content uniformity it will be useful to express results with respect to theoretical drug content.
- Authors should discuss effect of HPMC on dissolution rate of itraconazole. It is obvious that dissolution curve of HPMC films with nanosuspension reach constantly higher values compared to bulk nanosuspension. It will be good to present some experimental or theoretical data regarding effect of HPMC on itraconazole solubility and dissolution rate, and/or to perform dissolution testing of HPMC films with itraconazole (micro)suspension. Authors should also mention in methodology whether samples taken during dissolution testing were filtered and the filter pores size.
Author Response
Comment: This is well written paper describing development of oral films for improving dissolution rate and bioavailability of poorly soluble drug itraconazole. These are suggestions for authors to improve the quality of the paper.
Answer: Thank you very much for the positive comment and the suggestions.
Comment: It will be useful for reader to see photos and/or polarizing micrographs of prepared films (if films are sufficiently transparent) instead of only description of film appearance. Microscopy should proved absence of macroscopic aggregates of itraconazole crystals.
Answer: We agree that the photos would have increased the value of the appearance discussion and cleared out the table information. We have used microscope in the laboratory to help the visual analysis of the films in our studies. But unfortunately the photos are not high enough quality for publications.
Comment:Did authors used disks in pharmacopoeial disintegration apparatus? It is reasonable to expect that without disks films will floated out of the cylinders during test, particularly if the vessel is filled with medium, as described in this study.
Answer: Discs were not used. It is true that the floating might be problem in the disintegration testing without discs. But with discs the problem is that the films can be attached to the discs, which also affects the results. Based on the preliminary testing, we ended up not to use the discs in this study. Due to the problems in pharmacopoeial disintegration apparatus, we also performed the other disintegration test with another method. We have now added to the manuscript that the pharmacopoeial disintegration test was performed without the discs.
Comment:For content uniformity it will be useful to express results with respect to theoretical drug content.
Answer:thank you a lot for the excellent suggestion. We have changed the value to the Table 4 to be presented in mgs. (“Drug content uniformity (mg) 1.25±0.05”). We have also added to the discussion part a sentence where the determined drug amounts are correlated to the theoretical value: “the determined drug amount in the film pieces was 1.25±0.05 mg, while the theoretically calculated value was 1.2 mg.”
Comment:Authors should discuss effect of HPMC on dissolution rate of itraconazole. It is obvious that dissolution curve of HPMC films with nanosuspension reach constantly higher values compared to bulk nanosuspension. It will be good to present some experimental or theoretical data regarding effect of HPMC on itraconazole solubility and dissolution rate, and/or to perform dissolution testing of HPMC films with itraconazole (micro)suspension. Authors should also mention in methodology whether samples taken during dissolution testing were filtered and the filter pores size.
Answer: The samples were filtered with 0.45 μm PVDF filter before analysis; this information has been added to the text.
The dissolution from the films was almost in the same level (a little bit lower level) as from the nanosuspensions, indicating that the film formation didn’t destroy the nanoparticle structures. We have revised the text related to the dissolution test in the discussion part of the manuscript and added also some discussion of the role of the HPMC in the formulation. “The drug dissolution from the films was almost as fast as from the nanosuspension indicating good performance of the film formulation. The fast and complete drug release from itraconazole nanosuspensions is mainly due to the increased surface-to-volume ratio. Fast drug release of itraconazole from the films was attributed to the high surface area of drug nanocrystals available for wetting after fast disintegration of water-soluble HPMC films. HPMC has shown in earlier studies to be able to maintain the supersaturated state longer [15]. However, in this study the dissolution tests were performed under sink conditions, which meant that in the end point after the dissolution no supersaturated state was reached.”
Reviewer 2 Report
The paper entitled “Production of Itraconazole Nanocrystal Based Polymeric Film Formulations for Immediate Drug Release” is an interesting article.
In this articles the authors proposed the production of fast dissolving oral polymeric film formulations based on itraconazole nanocrystals .
Drug nanocrystals were prepared by wet pearl milling technique. In polymeric film formulations, hydroxypropyl methyl cellulose (HPMC) was used as a film forming polymer and glycerin as a plasticizer.
Nanocrystal suspensions and film formulations, thorough physicochemical characterization was realized including particle sizing and size deviation, film appearance, weight variation, thickness, folding endurance, drug content uniformity, disintegration time and dissolution profile. Film casting method represents a valid option for the production of itraconazole nanocrystal based with immediate release formulations.
The manuscript is consistence and accurate and well written. The data are presented and explain accurately.
The paper is interesting, a lot of characterizations are presented in this work. I have no hesitation to suggest the publication of this manuscript. The manuscript will be published in Pharmaceutics Journal after minor revision.
Specific comments:
Introduction section-paragraph 1: The authors are invited to describe better the novelty and the final application sector.
Film casting- paragraph 2.4.: The authors are invited to describe better this paragraph adding the quantities of different materials used during the process.
Author Response
Comment:The paper entitled “Production of Itraconazole Nanocrystal Based Polymeric Film Formulations for Immediate Drug Release” is an interesting article.
In this articles the authors proposed the production of fast dissolving oral polymeric film formulations based on itraconazole nanocrystals .
Drug nanocrystals were prepared by wet pearl milling technique. In polymeric film formulations, hydroxypropyl methyl cellulose (HPMC) was used as a film forming polymer and glycerin as a plasticizer.
Nanocrystal suspensions and film formulations, thorough physicochemical characterization was realized including particle sizing and size deviation, film appearance, weight variation, thickness, folding endurance, drug content uniformity, disintegration time and dissolution profile. Film casting method represents a valid option for the production of itraconazole nanocrystal based with immediate release formulations.
The manuscript is consistence and accurate and well written. The data are presented and explain accurately.
The paper is interesting, a lot of characterizations are presented in this work. I have no hesitation to suggest the publication of this manuscript. The manuscript will be published in Pharmaceutics Journal after minor revision.
Answer: Thank you very much for the positive comment!
Specific comments:
Comment:Introduction section-paragraph 1: The authors are invited to describe better the novelty and the final application sector.
Answer:thank you for the comment. We have sharpen the novelty and aims of the study part of the manuscript. We have added a sentence explaining the applicability and benefits of polymeric films to the first paragraph (“For nanocrystal based drug delivery systems polymeric films are especially beneficial, while no extra drying step is required: nanosuspensions can be added as such to the polymeric solution for the film casting or 3D printing.”)
And the last paragraph explaining the aims of the study has been changed in order to emphasize and clear out further the novelty and applicability of the study: “In this study, we aimed for production of immediate release formulation based on drug nanocrystals. Polymeric films were selected as the formulation due to their simple structure, good performance and fast production. Further functionality were build in to the formulation by using HPMC as a film former: HPMC stabilizes the supersaturated state, which is reached after the dissolution of drug nanocrystals. Presence of HPMC hinders the uncontrolled precipitation, which is often induced by the supersaturation. Accordingly, in this study, itraconazole nanocrystal based oral polymeric film formulations for immediate release purposes were produced. The film formulations were made by film casting method. HPMC was used as a film forming polymer. For film casting, the drug was nanomilled to nanocrystals. After the production of nanocrystal based polymeric film formulations, a thorough physicochemical and mechanical analysis (particle size and size deviation, film appearance, weight variation, thickness, folding endurance, drug content uniformity, disintegration time and dissolution profile) were performed in order to show their suitability for immediately drug release purposes.”
Comment:Film casting- paragraph 2.4.: The authors are invited to describe better this paragraph adding the quantities of different materials used during the process.
Answer: Thank you for the good suggestion. We have added more details to the film casting section and reference to the tables, where the exact compositions are presented later. “The amount of HPMC varied from 2.5 g to 5.3 g (in 100 mL total volume). The amount of glycerol was in all the studies 25%(w/w) as compared to the amount of HPMC (the exact compositions of the films are presented in Chapters 3.1 and 3.2 in Tables 1 and 3).”
The language has been proof-read in order to improve the language.